# Environmental Risk Assessment of Recombinant Viral Vector Vaccines against SARS-Cov-2

**DOI:** 10.3390/vaccines9050453

**Published:** 2021-05-03

**Authors:** Aline Baldo, Amaya Leunda, Nicolas Willemarck, Katia Pauwels

**Affiliations:** Sciensano, Service Biosafety and Biotechnology, Rue Juliette Wytsmanstraat 14, B-1050 Brussels, Belgium; Amaya.LeundaCasi@sciensano.be (A.L.); Nicolas.Willemarck@sciensano.be (N.W.); Katia.Pauwels@sciensano.be (K.P.)

**Keywords:** SARS-CoV-2, COVID-19, recombinant viral vector vaccines, environmental risk assessment, vaccination, biosafety

## Abstract

Severe acute respiratory syndrome coronavirus 2 (SARS-CoV-2) is the causative agent of the coronavirus disease 2019 (COVID-19) pandemic. Over the past months, considerable efforts have been put into developing effective and safe drugs and vaccines against SARS-CoV-2. Various platforms are being used for the development of COVID-19 vaccine candidates: recombinant viral vectors, protein-based vaccines, nucleic acid-based vaccines, and inactivated/attenuated virus. Recombinant viral vector vaccine candidates represent a significant part of those vaccine candidates in clinical development, with two already authorised for use in the European Union and one currently under rolling review by the European Medicines Agency (EMA). Since recombinant viral vector vaccine candidates are considered as genetically modified organisms (GMOs), their regulatory oversight includes besides an assessment of their quality, safety and efficacy, also an environmental risk assessment (ERA). The present article highlights the main characteristics of recombinant viral vector vaccine (candidates) against SARS-CoV-2 in the pipeline and discusses their features from an environmental risk point of view.

## 1. Introduction

The severe acute respiratory syndrome coronavirus 2 (SARS-CoV-2), the causative agent of the coronavirus disease 2019 (COVID-19) pandemic has caused millions of deaths worldwide and economic and social chaos internationally [1,2]. Vaccines are considered as an essential tool to prevent further morbidity and mortality [2]. Many efforts have been directed towards the rapid development of effective and safe COVID-19 vaccine candidates by use of a range of vaccine platforms. Currently, four vaccines are already authorised for use in the European Union (EU): two nucleic acid-based vaccines, BNT162b2 (Cominarty, Pfizer/BioNTech) and mRNA-1273 (COVID-19 Vaccine Moderna), and two recombinant adenoviral vector vaccines, ChAdOx1-S (COVID-19 Vaccine AstraZeneca) and Ad26.CoV2.S (COVID-19 Vaccine Janssen).

In the EU, the conduct of clinical trials with vaccines and marketing approval of such vaccines shall be in accordance with the provisions of Directive 2001/20/CE [3] and Regulation (EC) N° 726/2004 [4], respectively. The main objective of these regulations is to ensure that vaccines comply with relevant requirements in regard to their efficacy and safety for the involved human subjects and their quality control. Vaccines based on recombinant viral vectors are subject to additional regulatory requirements, as they are considered genetically modified organisms (GMOs). Some of these requirements, which have their legal basis in Directive 2001/18/EC [5], aim to assess aspects related to potential risks for human health and the environment, including animals, plants and micro-organisms, what is called the “environmental risk assessment (ERA)”.

In this article we present the general principles of the ERA and we elaborate on key features in relation to several recombinant viral vector COVID-19 vaccine candidates at various stages of clinical development or already approved for marketing in the EU (Table 1).

## 2. Environmental Risk Assessment

An ERA consists in the identification and characterisation of potential hazards associated with the GMO (in this case the recombinant viral vector vaccine) on human health (with focus on individuals other than patients or vaccinees) and the environment at large including animals, plants and micro-organisms, as well as an estimate of their probability of occurrence under the conditions of use. The risk to human health and the environment posed by each identified hazard of the GMO is estimated by combining the probability of its occurrence and the magnitude of its consequences. An overall risk is then determined by combining all of the individual risks [21,22]. The ERA is based on the weight of evidence methodology encompassing both qualitative and quantitative considerations [23] and is described using qualitative terms ranging from high, moderate, low to negligible [24]. After overall risk determination it is examined whether risk management measures need to be implemented in order to minimise the likelihood of adverse effects occurring. If no adverse effects were identified risk management strategies are not necessary. Furthermore, it is important to note that the ERA is conducted on a case-by-case basis.

When a recombinant viral vector is used, the ERA should take into account the characteristics of the viral vector backbone and the properties of the inserted gene sequence(s) and of the gene product(s).

### 2.1. Assessment of the Viral Vector Backbone

Potential harmful effects will vary depending on the viral vector backbone. Aspects such as host range, tissue tropism, potential of insertional mutagenesis in the host genome, reassortment, reconversion to virulence or pathogenicity by complementation events between the viral vector and circulating complementing viruses should be addressed as well as recombination events that may give rise to novel and uncharacterised viruses with reacquired pathogenicity or change in tissue tropism and host range. For the purpose of this paper, Table 2 gives a concise summary of how ERA could be applied to viral vector based vaccines against SARS-CoV-2.

As part of the assessment of the likelihood of occurrence of identified potential adverse effects, the exposure pathways through which the viral vector may interact with humans (persons other than those receiving the viral vector vaccine candidate), or the environment are to be considered. Exposure pathways are not adverse events per se, but rather mechanisms by which an adverse effect may occur. These exposure pathways include the biodistribution (dissemination in the host tissues), the dissemination occurring at the site of administration or during manipulation of the vaccine, the capacity to be transmitted by arthropods (cfr. vector-borne viruses) and the shedding (dissemination by means of excreta) [25].

For example, the evaluation of adverse events associated with shedding should consider the capacity for functional viral particles to retain their infectivity in the environment, the route of transmission, the capacity of the viral vector to infect cells of other persons or animals and the potentially adverse effects observed in humans and/or animals. However, shedding-based transmission to third parties is barely documented by experimental data. Therefore, the potential risk for the human population at large for animals due to shedding-based transmission is often only assessed on the basis of a weight of evidence of elements contributing to or involved in successful transmission.

Exposure to the recombinant viral vector may also occur at different steps of the manipulation of the vaccine, such as its manufacturing, its preparation or administration or the management of spills and waste disposal. Direct exposure may also result from accidental inoculation during the administration, via droplets or aerosols in contact with mucous membranes or as a result of injury due to the use of sharps. These exposure pathways can be drastically reduced by application of appropriate risk management measures.

### 2.2. Assessment of the Characteristics of the Inserted Gene Sequences

Assessing the potential impact of the inserted genetic material is another important aspect of the ERA because the inserted gene sequence(s) and/or the gene product(s) may affect the properties of the viral vector backbone or may have intrinsic hazardous (e.g., toxic, allergenic, oncogenic) properties (Table 2). Gene products that may be considered as potentially hazardous are detailed in the review of Bergmans et al. [26].

The recombinant viral vector COVID-19 vaccines discussed in this article all carry sequences that are derived from the Spike (S) protein of SARS-CoV-2. This protein forms homotrimers protruding from the viral surface [27] and mediates entry of the virus into the host cells. It is the main target of neutralising antibodies elicited upon COVID-19 infection and is understandably considered as the most important target antigen for vaccine development [28]. More particularly, viral entry is mediated by binding of the receptor-binding domain (RBD) in the S1 subunit of the S protein to the angiotensin-converting enzyme 2 (ACE2) of the host receptor. Subsequent fusion of the viral and host membranes involves the S2 subunit of the protein. Despite its key role in pathogenesis, the S protein appears not to have intrinsic hazardous properties as shown by numerous preclinical studies demonstrating immunogenicity and safety of recombinant viral vectors carrying sequences of the S protein [14,29,30].

Some bivalent vaccine candidates also use sequences coding for the SARS-CoV-2 nucleocapsid protein (N protein) [11,31] in addition to the S protein sequence. The N protein has been shown to be highly immunogenic, conserved among other coronaviruses and is found more stable over time as opposed to the S protein, rendering the N protein a potential good antigen candidate for vaccine against SARS-CoV-2 and its upcoming mutant strains/variants [32,33]. The N protein is a multifunctional protein, binding to viral RNA inside the virion, facilitating RNA replication and virus particle assembly and release. In host cells, N proteins have been shown to cause deregulation of the cell-cycle, to inhibit the interferon immune response and to induce apoptosis [34], hence conferring potential hazardous properties to the SARS-CoV-2 N protein. In SARS-CoV-2 vaccine candidate hAd5-S-Fusion+N-ET SD, an Enhanced T cell Stimulation Domain (ETSD) sequence is added to direct N protein to the endosomal-lysosomal subcellular compartment after translation [33]. This strategy aims to optimise N protein presentation for T helper cell activation. At the same time this delocalisation of the N protein might mitigate potential deleterious effects, as suggested by the results of preclinical studies with this vaccine in mice and non-human primates (NHP) [11,33].

In addition to the intrinsic characteristics of the inserted sequences, the ERA should also take into consideration the potential impact of strategies used for optimal design of antigen expression. These strategies aim at increasing genetic stability of the expressed transgene and/or inducing more effective immune responses in vaccinees. Approaches include codon-optimisation with the aim to increase transgene expression [35] or structure guided amino acid modification. The latter is exemplified by the modifications to stabilise the prefusion conformation of the S protein, so as to induce effective neutralising antibody responses against the prefusion spike and thus preventing ACE2 binding and cell entry. The ERA should consider the potential impact of altered nucleic acid sequences or altered amino acids in the S protein on the biodistribution or host range profile of recombinant viral vectors, in particular when the protein is expressed on the surface of the virion [36]. Indeed, the S protein appears to be a main determinant for cross-species infection events and is thought to play a role in host tropism, thereby highlighting its role for ERA considerations when it is also expressed on the surface of the recombinant viral vector [37].

As with any other virus, viral vectors may interact with other viruses present in the host and exchange genetic material with viral sequences that present high degree of homology. In theory, the insertion of exogenous viral sequences in the viral vector may extend the range of viruses with which recombination is possible. There are currently four other different low-pathogenic coronaviruses endemic in humans (HCoV-OC43, HCoV-HKU1, HCoV-NL63, and HCoV-229E) [28]. Therefore, the possibility of viral vectored COVID-19 vaccine to coinfect and colocalise in a same cell with endemic coronaviruses cannot be excluded. Whether this will lead to recombination events and result in the formation of novel uncharacterised chimeric viruses depends on several factors, including the replication capacity of the viral vector and the size of the insert so as to enable homologous recombination. In this respect, the insertion of only part of the sequence, e.g., the RBD domain of the S protein, or the insertion of a synthetic codon-optimised sequence may decrease the probability of homologous recombination.

## 3. Environmental Risk Assessment of Recombinant Viral Vector Vaccines against SARS-CoV-2

### 3.1. Replication Deficient Viral Vectors

#### 3.1.1. Adenoviral Vectors

Adenoviral vector vaccines belong to one of the best studied and most utilised vector platforms. It is therefore not surprising that they are used in the most advanced viral vector COVID-19 vaccines developed so far (Table 1) [38]. The COVID-19 vaccine candidates are based on human or simian adenoviral vectors and aim at eliciting a protective immune response of the recipient by delivering to cells the sequence of the antigenic SARS-CoV-2 S protein alone or combined with the N protein sequence.

Adenoviruses are non-enveloped double stranded DNA viruses. Human adenoviruses and some animal adenoviruses (monkeys, etc.) belong to the genus *Mastadenovirus* in the *Adenoviridae* family. They are classified into seven subgroups (A–G) and further into over 100 serotypes, more than half of which are known to infect humans. Human adenovirus serotype 5 (hAd5) of subgroup C is by far the most common circulating adenovirus with high seroprevalence rates in the worldwide population. Depending on the serotype, adenoviruses usually cause self-limiting mild disease affecting the respiratory tract (e.g., hAd5), eyes (e.g., hAd26) or the gastrointestinal tract (e.g., hAd40) in both humans and animals [39,40]. Most adenoviral serotypes are very stable in the environment and can persist for months on dry surfaces and for weeks in water. Adenoviruses are resistant to lipid disinfectants but are inactivated by formaldehyde, chlorine or alcohol-based disinfectants and by heating to 56 °C for 30 min.

Human adenoviruses (principally hAd5) have been studied for many years as the basis of adenoviral vector based vaccines. More recently non-human adenoviruses were also investigated as a vector tool in humans. Although uncommon, infections of humans by non-human adenovirus serotypes are occurring, which is mainly explained by structural and genomic similarities of viruses belonging to the same subgroup and because of broad tissue tropism of adenoviruses [41]. This feature has encouraged researchers to investigate the use of simian adenoviruses such as the chimpanzee ChAdY25 and the gorilla GRAd32 as vectors for human vaccines.

Adenoviruses and their derived vectors exhibit a broad tropism, infecting a variety of dividing and nondividing cells. They stay as episomes in host cell nuclei with an integration into the cell genome being an extremely rare event [42,43]. For safety and efficacy reasons, adenoviral vector based vaccines have been rendered replication-defective by the deletion of the entire or part of the early gene E1, thereby affecting the capacity of the vector to replicate but not its ability to transduce host cells and to serve as a gene delivery tool. The pathogenicity of the adenoviral vector is therefore significantly reduced. Usually, the transgene cassette is inserted in the E1 region of the vector. Moreover, the adenoviral early gene E3 known to inhibit host immunological pathways is also deleted, thereby increasing the transgene length capacity of the vector. Loss of immune evasion function of adenoviral vectors may result in a more effective clearance in the host and in an increased acute response such as inflammation [21]. In a second adenoviral vector generation used by ImmunityBio Inc. in its vaccine candidate, the E2b gene is also deleted which removes expression of the DNA polymerase and decreases late genes expression. The E2b deletion leads to further increase in length capacity for the transgene in the vector and diminishes potential immune responses against vector proteins [33].

Along with the numerous studies conducted with adenoviral vector constructs developed as a vaccine candidate or for gene therapy, a relatively good understanding of the host range, cellular tropism (biodistribution) and potential for genome integration has been acquired [41,44]. This knowledge supports the position that sequences encoding the S and N proteins from SARS-CoV-2 are not expected to alter the transmission route, the host range nor confer any deleterious effect to the adenoviral vector in particular due to the fact that these proteins are not expressed on the virion surface, as it happens in other types of COVID-19 vaccine candidates.

The impact of pre-existing immunity against common adenoviral serotypes is a major concern for their use in vaccines because it might attenuate the specific transgene-induced response. However, several strategies have been implemented in the context of adenoviral vector based vaccines against SARS-CoV-2 to overcome this problem. They include using a low prevalent human Ad26 (subgroup D) [45], deploying simian adenoviral vectors, such as ChAdY25 (subgroup E) or GRAd32 (subgroup C) circulating in the Chimpanzee and the Gorilla, respectively [12,46] or adopting a prime-boost heterologous vaccination with hAd26 and hAd5 as it is the case with the Sputnik vaccine [8] (Table 1). Altogether, features of human and simian adenoviral based vectors make them promising vaccine candidates for transient expression of the SARS-CoV-2 S protein and N protein with minimal risk of genomic insertional mutagenesis.

As with other viral vectors, a potential hazard associated with adenoviral vector based vaccines is the reversion to replication competency following homologous recombination events. The probability of replication competent adenoviruses (RCA) emergence is high during the manufacturing of the adenoviral vector based vaccines in a packaging cell line (usually HEK293) due to the presence of sequences originating from hAd5 that not only can complement in trans, so as to allow manufacturing of the viral particles, but that also can result in recombination events and the formation of RCA. An approach to limit homologous recombination events during manufacturing is to further reduce homologue sequences between viral vector sequences and the complementing sequences of the E1 region of the packaging cell line [47]. Nevertheless, the absence of RCA should be demonstrated by the manufacturers for each batch release of the adenoviral vectored vaccine candidate as part of the routine quality control [21].

Emergence of RCA might also occur after administration of the vaccine by homologous recombination events between the adenoviral vector based vaccine sequences and the parental wild-type virus or other related human adenovirus infecting the same host cell. This remains a hypothetical hazard since no recombination events have been reported so far with replication-defective vectors. Upon coinfecting with wild-type virus such as hAd5, homologous recombination events might theoretically lead to the generation of replicative hAd5 variants harbouring the E1 gene without transgenes or, depending on the adenoviral construct, may result in replication competent simian Adenovirus/hAd5 or hAd26/hAd5 chimeric vectors without transgenes and replication-deficient chimeric vectors harbouring the transgene sequence. For example, considering bivalent vaccines with hAd26/hAd5 vectored prime-boost regimen, such as for Sputnik V, recombination events between hAd26 and hAd5, if they would occur, would ultimately result in vectors with E3 and E1 deleted versions.

Of note, simian adenoviral vectors (ChAdY25 and GRAd32) have been further genetically modified to harbour E4 coding regions from hAd5 [12,46] so as to optimise vector vaccines’ growth rate and yield in human packaging cells. Yet, from an ERA perspective, it increases sequence homologies and the propensity of homologous recombination with wild-type hAd5. On the other hand, chances of homologous recombination events between hAd26 or simian based adenoviral vectors and a wild-type hAd5 might be low as these viruses belong to different adenovirus subgroups and share short E1 sequence homology regions.

In the scenario that RCA and replication competent chimeric adenoviral vectors are shed by vaccinees, the general population and the environment might be exposed. Potential adverse effects on human health and the environment of replication competent chimeric viruses are unclear although a decreased replication capacity of adenovirus chimeras from different species has been reported [48].

For RCA and replication competent chimeric viruses’ emergence to occur, colocalisation of the adenoviral vector vaccines with another naturally occurring virus should take place in the same cell. Colocalisation is nevertheless an event expected to occur with low probability firstly because of the relatively short time presence of the adenoviral vector in infected cells resulting from its replication incompetence and because of its rapid clearance by the host immune response. The use of adenoviral vector vaccines derived from low prevalence adenoviruses such as hAd26 or simian adenoviruses (ChAdY25 and GRAd32) further decrease chances of a cell coinfection with their wild-type counterpart. ChAdY25 and GRAd32 only circulate among chimpanzees and gorilla, respectively, which are in most cases not present in our direct environment. This might not be the case in other parts of the world such as Africa and Asia.

Probability of colocalisation of vectors with wild-type viruses will also be influenced by the route of administration of the vaccine. Mucosal membranes of the respiratory tract, the eyes or the gastrointestinal tract are a predominant portal of entry for wild-type circulating adenoviruses. On the other side, most of the adenoviral vector based vaccines against SARS-CoV-2 are administrated by intramuscular injection (IM), often into the deltoid muscle (Table 1). According to studies in animals, hAd5 vector biodistribution after IM administration has been shown in liver, lung and spleen [49]. No adenoviral vectors have been detected in human patients’ excreta (stool, urine, throat swab) after IM administration in the leg, meaning that the natural portal of entry of wild-type viruses and the route of administration differ.

Other adenoviral vector based vaccines undergoing phase I or II clinical trials are intended to be administrated by subcutaneous and oral mode (Table 1). Whereas subcutaneous entry may not mimic natural infection of adenoviruses, the oral route of administration could be compared to a natural portal of entry through mucosal membranes of wild-type adenoviruses into organisms. For instance, for hAd5-based vaccine candidates intended to be administrated through buccal mucosa, the probability of coinfection of the same cell with wild-type adenovirus might be increased and hence might increase the likelihood of homologous recombination events as well.

It has been shown that shedding of adenovirus vectors and duration of shedding will depend on the vector serotype, the administrated dose as well as the mode of administration [25,49]. Based on the existing literature [25,44,49], shedding following IM administration of adenoviral vectors in humans has been reported as a very rare event. Viral particles have been detected at the site of injection shortly after vaccination. As far as we know, shedding of adenoviral vectors after oral or subcutaneous administrations has not been investigated in humans.

Another potential safety issue to be considered associated with the biodistribution and shedding of adenoviral vector based vaccines is the dissemination of the vector to the gonads, resulting in the risk of germ-line transmission. Despite expression of Coxsackievirus and Adenovirus Receptor (CAR) mediating adenovirus cell entry on mouse germ cells, studies on systemic administration of hAd5 vectors [50], and testicular spermatids and epididymal sperm analysis did not show any evidence for infection. The same results were observed when inoculating mouse ovaries and oocytes directly with hAd5 vectors.

Overall, data currently available on tropism and biodistribution, potential for recombination and shedding show that the SARS-CoV-2 vaccine candidates based on adenoviral vectors listed in Table 1 have a good safety profile in regard to their potential risks for human health and the environment. It should also be noted that other vaccines based on adenoviral vectors derived from different simian or human adenovirus have already been tested in several clinical studies without ERA-related concerns having been notified. Specific attention should be given to the adenoviral vector based vaccines against SARS-CoV-2 administered by oral or subcutaneous mode since the proposed route of administration might increase the probability of vector shedding or coinfection of a same cell with vaccine and a naturally occurring adenovirus. From the ERA viewpoint, it would be interesting to monitor RCA emergence in the ongoing clinical trials with these vaccine candidates in humans, and to conduct shedding studies as part of clinical development plan.

#### 3.1.2. Modified Vaccinia Virus Ankara (MVA) Vectors

Modified Vaccinia Virus Ankara (MVA) viral vectors, which are derived from an orthopoxvirus strain developed in the 1970s as a vaccine against smallpox, are another type of replication deficient recombinant vectors widely tested for vaccination or gene therapy applications. As opposed to the adenoviral vectors described above, MVA remains localised in the cytoplasm [51] thereby alleviating concerns associated with integration in the host genome [52]. Interestingly, an increasing number of applications include the use of adenoviral vector constructs as prime vaccine in combination with MVA derived viral vectors to circumvent a boost of neutralising antibodies against the viral backbone. Building on the good safety and immunogenic profile demonstrated through various clinical studies including infants and immunocompromised patients, MVA viral vectors are understandably seen as a promising vector vaccine platform against infectious diseases [53,54,55]. More recently, a recombinant MVA expressing the S protein of the Middle East respiratory syndrome coronavirus (MERS-CoV), a close relative to SARS-CoV-2, revealed safety and immunogenicity in a first-in-human phase I clinical study [56] and a viral vector vaccine named MVA-SARS-2-S is currently being investigated in a phase I clinical trial [30]. Another vaccine candidate based on a fully synthetic form of MVA (sMVA) is currently tested in a phase I clinical trial [57]. The vaccine platform based on sMVA has been developed to rapidly produce sMVA vectors and to concomitantly combine two antigen sequences in a single MVA vector [15]. This recombinant sMVA vaccine candidate expresses SARS-CoV-2 S protein and N protein [57]. The recombinant MVA vectors generated from chemically synthesised DNA have the same characteristics in vitro and in vivo as compared with those of wild-type MVA in mice [15].

Several features of recombinant MVA vectors from an environmental risk perspective have already been reviewed [58,59]. Issues of particular importance during the ERA of MVA vectors are the homogeneity and genetic stability of the recombinant MVA vector and the potential for recombination and reconversion to the wild-type.

Information regarding the homogeneity of the MVA strain is important in order to exclude the presence of replication competent MVA particles. MVA-SARS-2-S has been developed using the MVA platform technology based on the MVA strain “F6 from Ludwig Maximilians-Universitat (LMU) München”, for which clonal genetic homogeneity has been confirmed by analysis of viral DNA [60]. Moreover, genetic identity and the in vivo genetic stability of the recombinant MVA-SARS-2-S was confirmed during the pre-clinical study [14].

Regarding recombination and reconversion to the wild-type, it should be noted that MVA is a highly attenuated orthopoxvirus adapted to avian cells, which has lost its ability to replicate in mammalian hosts. The attenuation of MVA is based on 570 serial passages in primary chicken embryo fibroblasts (CEFs), resulting in a genomic loss of approximatively 15% compared to the parental Chorioallantoic vaccinia Ankara (CVA) virus strain and reducing not only its virulence and pathogenesis [58,60] but also the risk for reconversion to the wild type. However, it has been suggested that some of the disrupted or deleted genes could be rescued by recombination in case of coinfection of a MVA-based vaccine and a naturally occurring orthopoxvirus (OPV) [61]. Such an event, however, is considered very unlikely because there are no known human poxviruses [58].

Preclinical studies in BALB/C mice with MVA-SARS-2-S, which harbours the sequences for the full-length S protein of SARS CoV-2, revealed no toxicity concerns upon IM injection of the construct [14]. Other data relevant to the ERA such as biodistribution and shedding of MVA-SARS-2-S are currently missing. However, information on the biodistribution of the same vector expressing S protein from MERS-Co-V reveals that the viral DNA remained restricted to the parental site and in draining lymph nodes after IM administration in mice. MVA DNA was not found in other peripheral organs, with the exception of very low numbers of copies detected in single samples and was not found in secretions (urine and faeces) [62]. A particular feature associated with poxviruses in terms of potential dissemination and hence exposure pathway for human population and the environment involves the formation of skin pock lesions. In this regard, the inoculation of MVA-SARS-2-S via IM route minimises or even abolishes the development of skin pock lesions on the administration site, thereby reducing the risk of dissemination via the site of administration [58].

Overall, while MVA shows high environmental stability and high resistance to desiccation, as for all poxviruses, the environmental impact during unintended environmental exposure might be limited because MVA is not able to replicate in mammalian host, vaccinia virus has no natural reservoir [63] and recombination events in human hosts are unlikely as no poxviruses circulate among humans.

#### 3.1.3. Recombinant Influenza Virus Vectors

Though less commonly investigated compared to the viral vectors addressed above, the ability of influenza to carry exogenous viral sequences of other viruses make this backbone another attractive option for developing viral vectored vaccines against influenza strains or other viral diseases of interest such as COVID-19. Influenza viruses are members of the *Orthomyxoviridae* family of segmented, negative sense single-stranded RNA viruses and are mostly subdivided on the basis of the antigenic nature of their membrane-bound surface glycoproteins, haemagglutinin (HA) and neuraminidase (NA).

Several influenza-based vaccines against COVID-19 are currently being investigated for intranasal application, with DelNS1-2019-nCoV-RBD-OPT1 as the only one currently in clinical phase [16,17] (Table 1). DelNS1-2019-nCoV-RBD-OPT1 is a live attenuated influenza virus due to the deletion of the nonstructural protein 1 (NS1), a key virulence element with multifunctional roles in virus replication and a potent antagonist of host immune response [16,17] through regulating the splicing of the M gene in M2 [64]. Though limited information is currently available, it is clear that CoroFlu, another influenza vector currently tested in preclinical studies, is derived from a self-limiting version of the influenza virus, so called M2SR, which is restricted to one single round of infection as a result of a deletion of a portion of the M2 gene [65]. These attenuated vectors are produced using cell-cultures which harbour sequences that can complement the reduced replication profile. Of note, this cell-culture based manufacturing process, as opposed to the egg-based manufacturing of common influenza vaccines, offers some interesting prospects because egg-allergies to the flu vaccine for human use as well as concerns with manufacturing shortage in case of avian influenza pandemics are alleviated.

Different clinical trials involving intranasal administration and encompassing hundreds of subjects have shown the backbone to be safe and well tolerated [66]. The intranasal mode of administration is considered to mimic a natural route of infection and is therefore thought to possibly trigger a higher immunogenicity as compared to less natural routes such as intramuscular injection. From a vaccine safety perspective, and similar to adenoviral vectored vectors, it is noticed that influenza-based vectors are likely to induce an immune response against backbone associated antigens, such as the hemagglutinin protein.

No shedding data of the influenza viral vectored COVID-19 vaccine candidates are available yet, but at least transient local shedding is to be expected. Although influenza is not remarkably persistent outside the host or resistant to disinfectants or physical inactivation (it can be rapidly inactivated by heating at 56 °C or with commonly used disinfectants), it can remain infectious for days in humid conditions. Transmission, in addition to droplets, occurs predominantly via contact of the mucous membranes.

An important aspect to be considered from an ERA point of view with viral vectors derived from influenza virus are possible reassortment events between circulating wild-type strains and the used vaccine strain. This particularly holds true when the vaccine is administered by a naturally occurring portal of entry of influenza viruses. Intranasal administration of the vaccine candidates is likely to increase the propensity of coinfection in the same cell with a complementing virus. Should reassortment occur, it needs to be assessed whether the resulting reassorted vaccine candidates confer adverse effects which could be of more, the same or less concern compared to the circulating strains [67].

The M2SR and DelNS1 backbones have been shown to be genetically stable over multiple passages and reversion to virulence has not been reported to date [68,69]. However, it is unclear whether experiments on coculturing with other influenza viruses have been conducted. As the vaccines are nonreplicative in humans, the probability to reassort with replicative influenza virus is limited within the decay period (of days) of the vaccine. On the other hand, some in vitro studies indicate that blocking the M2 gene alone is not effective enough to prevent replication [67,70]. Hence, more data are needed to better understand the likelihood of exchange of genetic material with or the complementation by coinfecting wild-type influenza strains. Considering that Influenza A virus can remain infective for days in humid conditions, and has many permissive hosts, such as wild birds and mammals (pigs, horses, seals, cats, ferrets, minks), a precautionary approach is justified when implementing risk management measures.

### 3.2. Replication Competent Viral Vectors

#### 3.2.1. Live-Attenuated Measles Virus Vector

Unlike all of the viral vectors described above, measles virus vectors are replicating recombinant viral vectors derived from the live attenuated measles Schwarz strain (MV-Schwarz). Building on the common endeavour of viral vector technology to express heterologous viral antigens, MV-Schwartz based vectors stably express large, heterologous antigen-coding sequences up to 6 kb long sentence [71,72].

The measles vaccine is a live-attenuated negative-stranded RNA virus, which induces life-long protective immunity after a single injection and has dramatically reduced childhood mortality from measles by 90% since its introduction. Given its proven safety record, with millions of vaccine doses safely administrated over more than 50 years of use [73], it is understandable that it has also been explored as a viral vector for a measles virus-derived COVID-19 vaccine candidate, in this case V591, that carries the sequences encoding for SARS-CoV-2 S protein and entered phase I clinical trial [18] (Table 1).

Environmental risks related to the use of recombinant measles vectors as vaccines have been reviewed [74]. Data that can inform on ERA of this viral vector have been obtained throughout the numerous recombinant measles vaccines developed against several viral pathogens including coronaviruses SARS-CoV [75,76] and MERS-CoV [77,78] that have so far been generated and tested in animal models [79]. Moreover, recombinant attenuated MV Schwarz strains are currently being tested in several clinical trials as vaccine against HIV [80], Chikungunya [81] and Zika virus [82]. The vaccine candidate for prevention of Chikungunya virus, the most advanced vaccine candidate in development [81,83], has been evaluated in several phase I and phase II trials and was consistently found to be safe and well tolerated.

Data point to the remarkably stability of the MV genome as compared to other RNA viruses such as influenza and HIV. MV vaccine strains are characterised by relative high genetic stability even after prolonged replication in the human host [84]. This finding seems to corroborate with the observation that reversion to pathogenicity of measles vaccine strains and subsequent transmission to other individuals have not been reported to date [73,74].

The MV genome also exhibits interesting features with respect to the likelihood of recombination. The only known reservoir of MV is human [85,86]. Even if non-human primates can be infected, the overall population is estimated to be too low to maintain transmission [87]. No recombinant viruses have been isolated from natural infections and there has been no conclusive evidence to date of any genetic recombination events between MV vaccine strains and wild-type strains in humans coinfected with both viruses [88], thereby corroborating the position that recombination does not occur in paramyxoviruses. Consequently, the risk of recombination between V591 and wild-type MV is considered negligible.

Furthermore, revaccination of individuals already immunised with MV vectored vaccines have been shown to result in a boost of anti-MV antibodies, indicating that the live vaccine still can replicate despite pre-existing immunity [89,90,91].

MV-based vaccine candidates for Zika, Lassa, Dengue and Chikungunya are well tolerated, and no signs of systemic toxicity were noted. None of the heterologous antigens inserted into the measles vector changed the toxicity profile, biodistribution, shedding behaviour or tropism. Biodistribution of MV-based vaccines were similar to that of the parental MV-Schwarz vaccine strain.

Shedding of MV-CHIK was assessed using real-time PCR in urine and saliva samples from a subset of participants during the clinical study and MV-CHIK RNA was not detected in any of the samples analysed [83].

When addressing the possible consequences of individuals other than the vaccinees upon exposure to viral particles shed by the vaccinees, one should consider that most individuals in industrialised countries are immune to the wild-type measles virus as a result of natural infection or vaccination. This means that, should exposure occur, the immune response of most people would rapidly clear the measles vector construct, therefore greatly reducing the probability of dissemination by shedding. Moreover, the measles virus is not stable in the environment, retaining its infectivity for less than 2 h on surfaces or objects [92].

Results of phase a I clinical study showed that the vaccine candidate V591 was safe. However, the immune response was inferior to those seen following natural infection and those reported for other SARS-CoV-2 vaccine candidates [93]. The development of V591 has therefore been stopped.

#### 3.2.2. Vesicular Stomatitis Virus (VSV)-Vectors

Vesicular stomatitis virus (VSV)-based vectors are replicating viral vectors derived from a vector-borne virus which has several animal hosts and therefore exhibit some other relevant features from an environmental risk point of view [94]. VSV is a single-stranded negative sense RNA virus, belonging to the family *Rhabdoviridae*, genus *Vesiculovirus* and has eight main serotypes, of which serotype VSV-Indiana (VSV-I) and New Jersey (VSV-NJ) have been used as a vector backbone for the development of viral vectors. VSV replicates within the cytoplasm of infected cells and does not integrate into the cellular genome.

The life cycle of VSV involves sandflies and rodent reservoirs. VSV-NJ and VSV-I can be transmitted between livestock by direct contact, likely including droplet spread and fomites, as well as mechanically by non-biting houseflies and face flies [95,96]. Mechanical transmission by flies and animal-to-animal or animal-to-human transmission may occur through direct contact with vesicular lesions. Infection of humans with wild-type VSV (wt-VSV) can cause an influenza-like disease, usually without vesicle formation [95,97,98], but no documented evidence exists for human-to-human transmission or human-to-animal transmission of VSV [99].

VSV-related disease is significant in pigs, cattle, and horses and is predominantly reported in America [95,98,100,101,102]. Although causing crusting and vesiculation of the mucous membranes and skin and leading to significant economic losses to livestock farmers, VSV has been removed from the list by the World Organisation for Animal Health (OIE) as a reportable animal disease [103] because of the mild, self-limiting nature of the disease and unlikely international spread through trade of animals. Wild-type-VSV has also been reported to circulate in bats. Upon experimental infection with VSV-NJ or VSV-I, viremia has been demonstrated in deer mice (*Peromyscus maniculatus*), laboratory mice (*Mus musculus*), spiny rats (*Proechimys semispinosus*), and Syrian hamsters (*Mesocricetus auratus*).

VSV is remarkably resistant to extreme pH values in particular in the alkaline range but can be rapidly inactivated by heating at 55 °C or higher. The virus is highly sensitive to inactivation by commonly used disinfectants such as aldehydes, alcohols, and detergents.

The relatively low prevalence of immunity to the vector, its nonintegrating properties and the capacity for large payload render VSV as an attractive vector-platform. The approach in developing VSV vectors consists in deleting (part of) the sequence encoding for the natural occurring envelope protein responsible for attachment to cells, VSV G, and to replace it with sequences encoding one or more heterologous envelope proteins able to reconstitute the attachment, fusion and budding function.

HIV-1, hantaviruses, filoviruses, arenaviruses, and influenza viruses are pathogens for which VSV-vectored vaccines are in preclinical development [104,105,106,107,108] while other VSV-based vaccines against emerging RNA viruses are already in clinical use [109].

Noteworthy, rVSVΔG-ZEBOV-GP, a VSV-Ebola licensed human vaccine in which the VSV G gene has been replaced by the filovirus GP gene, is the first VSV-vectored vaccine for which a full ERA has been conducted as per EU regulatory requirements for the marketing authorisation of medicinal products containing or consisting of genetically modified organisms [4,110].

Building on the experience achieved with VSV-eGFP-SARSCoV-2, which displays S protein on its virion surface and serves as a SARS-CoV-2 surrogate in neutralisation assays [29], the potential of VSV-vectored vaccines against COVID-19 disease has been supported by data showing immunogenicity and in vivo efficacy of VSV-eGFP-SARS-CoV-2 upon intranasal route of administration in a mouse model of SARS-CoV-2 pathogenesis [111]. At the time of writing two VSV-vectored COVID-19 vaccines have entered clinical development [19,20] (Table 1).

Importantly, and unlike most other COVID-19 vaccines, VSV-vectored COVID-19 vaccines express the S protein on the surface of the virion. For virions expressing surface -exposed heterologous proteins playing a role in cell attachment, the collection of specific data on tropism, biodistribution and shedding properties becomes key to a good understanding of their in vivo behaviour. This is particularly true for replicating viral vectors so as to anticipate any potential adverse effects on human population and the environment upon release into the environment.

Of note, tropism and biodistribution properties of rVSVΔG-ZEBOV-GP do not feature exactly the same profile as its parental virus. rVSV-ZEBOV virus (and other rVSV-filoviruses) fails to replicate in Jurkat cells which are susceptible to wt-VSV but not to Ebola virus. On the other hand, the rVSVΔG-ZEBOV-GP was found to infect keratinocytes in humans, a feature of wt-VSV. These findings indicate that even with a foreign envelope protein, the rVSVΔG-ZEBOV-GP virus shares tropism with the wt-VSV [106,108,112]. Neurovirulence of rVSVΔG-ZEBOV-GP vaccine is markedly attenuated compared to wt- VSV [97,113,114]. However, when inoculated by the intracerebral route, the rVSVΔG-ZEBOV-GP vaccine is virulent only for newborn mice, while no clinical signs or significant histopathological lesions were observed in non-human primates inoculated by the intrathalamic route [97,106,108,112,115,116,117]. The latter shows that a better understanding of tropism and biodistribution patterns is crucial, particularly in animal models that may give a better predictability towards clinical translation.

No shedding data has yet been reported for VSV-vectored COVID-19 vaccines. Shedding data obtained from nonclinical studies in larger animals like non-human primates may be informative. However, due to the inherent limitation associated with animal models (e.g., difference in virus clearance), the absence of viral shedding in animal studies does not allow to conclude on absence of viral shedding in humans or cannot be used to waive the collection of shedding data in humans. For this reason, and unless shedding data are obtained, the possibility of person-to-person transmission should be considered in an ERA. On the basis of the biodistribution profile of VSV-vectored COVID-19 vaccine in large animals, the collection of saliva, urine and stool samples from clinical trial participants could be considered to investigate the presence of viral particles in these samples and to assess the likelihood of transmission to nonvaccinated individuals.

The route of administration of the vaccine should also be considered in light of a potential effect on biodistribution and shedding properties. While both VSV-vectored COVID-19 vaccine candidates currently under investigation are administered intramuscularly, an oral or intranasal VSV-vectored vaccine could be an effective immunisation strategy against SARS-CoV-2, as supported by the findings obtained with VSV-vectored mucosal vaccines against MERS-CoV [118]. Referring to considerations on the likelihood of coinfection with other wild-type viruses mentioned in previous sections, the likelihood of recombination, with other negative sense RNA viruses upon oral or intranasal administration of vaccine warrants careful consideration.

VSV-vectored COVID-19 vaccines have a single stranded RNA genome and replicate within the cytoplasm and recombination events with wt-VSV or coinfecting negative sense RNA-viruses cannot be excluded [119]. On the other hand, an assessment of the likelihood of recombination events with wt-VSV should also consider the geographic distribution of wt-VSV.

Along with collecting biodistribution and shedding data, consideration should also be given to the replication competence and viremia levels. Given that VSV is a vector-born virus, the likelihood that an insect may transmit the viral vector to another individual or animal upon a blood meal from an immunised person should be assessed. However, if replication capacity and detected viremia levels are comparable to levels obtained with rVSVDG-ZEBOV-GP, it may provide a justification to waive the collection of data obtained in relevant insect lines as conducted in the context of the ERA of rVSVDG-ZEBOV-GP [96].

As a conclusion, as long as data on biodistribution, shedding and viremia levels are not completed, due consideration is to be given to measures aiming at minimising contact of trial participants with immunocompromised individuals, vulnerable persons or persons who are at increased risk of severe COVID-19 disease. Regarding the risk for animals, and if shedding data is lacking, human-to-animal transmission can also not be excluded. Risk mitigation measures could be implemented such as exclusion criteria for participants who are likely to have contact with animals, in particular with cats, dogs, minks, pigs, horses or cattle.

## 4. Discussion

The current COVID-19 public health emergency context has further emphasised the importance of preparedness by the development of novel vaccines. Accelerated data assessment, while keeping high standards for evidence-based demonstration of safety, efficacy and quality, remains key in this process. Moreover, as SARS-CoV-2 variants continue to be identified and their virulence, infectiousness, transmissibility and ability to escape vaccine-associated protection remains to be determined, there might be a need to regularly update current COVID-19 vaccines to emerging variants. Most SARS-CoV-2 variants of concern emerged in the fall of 2020 with most notably the United Kingdom variant B.1.1.7, the South Africa variant B.1.351 and the Brazil variant P.1 [120]. From an ERA point of view, this raises the question to which extent data obtained from vaccine candidates that are in a more advanced phase of development could be extrapolated to support the ERA of novel vaccines against SARS-CoV-2 or its variant strains using the same viral vector platform. For example, data on biodistribution and shedding profiles obtained with constructs using a replication-incompetent viral vector in which the transgene is not altering the vector capsid could be considered sufficient to inform the ERA of novel constructs using the same viral backbone in terms of biodistribution profile, shedding profile and capacity to be transmitted to non-vaccinees or the environment.

The accelerated data assessment associated with the use of viral vector platform technology could be a means to address the need for preparedness when pandemics or public health emergencies of international concern emerge. This need has been identified not only by vaccine developers but is also supported by health authorities and organisations. As the collection of data for the ERA are often perceived as cumbersome by vaccine developers, and considering the context of the SARS-CoV-2 pandemic, the European Union (EU) adopted a regulation providing for a temporary derogation from European legislation on GMOs [121] with a twofold objective: (i) to support the development of safe and effective medicinal products for the treatment or prevention of COVID-19 by facilitating the possibility to conduct clinical trials on medicinal products containing or consisting of GMOs as soon as possible (ii) to ensure rapid availability of COVID-19 vaccines and treatments in case of emergency. This means that the conduct of clinical trials is temporarily exempted from an ERA. This regulation is temporary and shall apply as long as the World Health Organization (WHO) classifies COVID-19 as a pandemic or as long as an implementing decision is applicable by which the European Commission recognises a situation of public health emergency due to COVID-19.

As outlined in this paper, the ERA is both a case-by-case and a weight of evidence approach. It cannot be excluded that for some COVID-19 clinical trials, which are exempted from GMO legislation as per Regulation 2020/1043, uncertainties may remain with respect to potentially altered biodistribution, shedding, genetic stability and transmissibility of the recombinant viral vector upon insertion of another transgene. This is particularly true when the transgene is expressed on the surface of the viral particle, hence refraining from the possibility to extrapolate data from one construct to another. Moreover, ERA data may not have been collected yet for less well-studied viral vectors or, if ERA data are available, they may not be extrapolatable due to an altered and preferred mode of administration for the vaccine candidate to be investigated. In this context, and because remaining uncertainties with respect to the risk assessment can be handled by the implementation of risk management measures, it should be noted that Regulation 2020/1043 specifies that sponsors shall implement appropriate measures to minimise foreseeable negative environmental impacts resulting from the intended or unintended release of the investigational medicinal product into the environment.

Currently, the collection and assessment of data in the context of vaccine development is mainly triggered by dataset requirements that support safety, quality and efficacy assessment of the medicinal product and focus on the health safety aspect of the patient or the vaccinee themselves. However, a proper assessment of environmental safety aspects related to personnel handling the vaccine (occupational exposure), close contacts of the vaccinees and the environment (including animals, plants and micro-organisms) necessitates the collection of data on shedding, the person-to-person or person-to-animal transmissibility of the vaccine or its capacity to exchange genetic information with circulating viruses. The collection of such data is barely addressed in early phases of the vaccine development notwithstanding this data are crucial for the ERA of the vaccine candidate. It would therefore be beneficial to collect such experimental data as early as possible in the preclinical and clinical developmental plan of the vaccine. Not only could this data better inform the ERA associated to the use of the candidate-vaccine, but it could also greatly facilitate the assessment at the time of marketing application whenever public health emergencies are emerging and poses time challenges for gathering experimental data. 

## 5. Conclusions

Several viral vectors described in this article have been investigated for many years in the light of the development of vaccines. With the advent of the current pandemic, it becomes clear that these efforts have culminated in the rapid development of COVID-19 vaccine candidates. Along with the numerous studies focusing on the safety of the vaccine for the vaccinee, the quality and/or the efficacy, an adequate evaluation of data from an ERA point of view is of importance. Building on the experience gained with some viral platforms and/or the collection of data for other emerging viral vectors, the case-by-case principle as embedded in the ERA methodology and illustrated in this article should provide a solid basis to guarantee a scientifically sound, adequate and proportionate approach.

## Figures and Tables

**Table 1 vaccines-09-00453-t001:** Viral vector based vaccines against SARS-CoV-2 that are in clinical development or already authorised for use in the European Union.

Viral Vector Vaccine Candidate	COVID-19 Vaccine Developer/Manufacturer	Genetic Modifications of the Vector	Inserted Gene Sequences	Route of Administration	Clinical Stage	References
Ad5-nCov/Convidicea	CaniSino Biologicals Inc., Bejin Institute of Biotechnology CanSino Biologicals Inc/Institute of Biotechnology, Academy of Military Medical Sciences, PLA of China	Nonreplicating human Ad5 E1 and E3 deleted	Optimised Spike coding sequence	Intramuscular (IM) Mucosal administration	Phase III (NCT04526990), Phase I/II (NCT04552366)	[6,7]
Gam-COVID-Vac/Sputnik V COVID-19 vaccine	Gamaleya Research Institute	Nonreplicating human Ad26 and Ad5 E1 and E3 deleted	Full-length glycoprotein S	IM	Phase III (NCT04530396)	[8]
Ad26.CoV2.S/COVID-19 vaccine Janssen	Janssen Pharmaceutical Companies	Nonreplicating human Ad26 E1 and E3-deleted	Stabilised wt Spike protein in the prefusion conformation	IM	Authorised for use in the European Union (EU)	[9]
ChAdOx1- S/COVID-19 vaccine AstraZeneca	AstraZeneca—University of Oxford	Nonreplicating Chimpanzee adenovirus ChAdY25 E1 and E3-deleted exchange the native E4 orf4, orf6 and orf6/7 genes for those from human adenovirus hAd5	Codon-optimised full-length Spike protein of SARS-CoV-2	IM	Authorised for use in the EU	[10]
hAd5-S-Fusion+N-ET SD vaccine	ImmunityBio Inc.	Nonreplicating human Ad5 E1, E2b and E3 deleted	Full length Spike fusion protein and nucleocapsid with an enhanced T-cell stimulation domain	Subcutaneous and sublingual boost	NCT04591717 (Phase I)	[11]
GRAd-CoV-2	ReiThera srl Leukocare Univercells	Nonreplicating Gorilla Ad32	S protein	IM	NCT04528641 (Phase I)	[12]
VXA-CoV2-1	Vaxart	Nonreplicating human Ad5	S and N proteins and dsRNA	Oral tablet	NCT04563702 (Phase I)	[13]
MVA-SARS-2-S	Universitätsklinikum Hamburg-Eppendorf Ludwig-Maximilians-University of Munich	Modified vaccinia virus Ankara (MVA)	Full length S protein	IM	NCT04569383 (Phase I)	[14]
COH04S1	City of Hope Medical Center	A synthetic MVA	S and N proteins	IM	NCT04639466 (Phase I)	[15]
DelNS1-2019-nCoV-RBD-OPT1	Xiamen University, Beijing Wantai Biological Pharmacy	Influenza virus vector: deletion of NS1 gene	Receptor Binding Domain (RBD) of S protein	Intranasal spray	ChiCTR2000037782 (Phase I) ChiCTR2000039715 (Phase II)	[16,17]
V591 (TMV-083)	Institut Pasteur Themis Bioscience GmbH University of Pittsburg Merck	Measles virus Schwarz vaccine strain	S glycoprotein in its prefusion conformation	IM	NCT04497298 (Phase I)	[18]
V590	IAVI Merck (Merck Sharp & Dohme Corp.)	Vesicular stomatitis virus (VSV)	S protein	IM	NCT04569786 (Phase I)	[19]
R-VSV-SARS-CoV-2-S	Israel Institute for Biological research (IIBR)	Vesicular stomatitis virus (VSV)	S protein	IM	NCT04608305 (Phase II)	[20]

**Table 2 vaccines-09-00453-t002:** Elements related to viral vector based vaccines against SARS-CoV-2 to assess in context of an environmental risk assessment (ERA).

Elements of the ERA to be considered related to the backbone	- Intrinsic hazardous properties (e.g., pathogenicity, toxic, allergenic and oncogenic properties, e.g., risk of genomic insertions with transformative changes) - Reconversion abilities to wild-type features, including the likelihood of recombination, reassortment, reconversion or complementation events between the viral vector and circulating complementing viruses to an uncharacterised virus variant with reacquired pathogenicity or change in tissue tropism and host range - Dissemination abilities, due to exposure events (during production, administration), shedding, biodistribution (leading to possible vertical transmission, vector-borne transmission)
Elements of the ERA to be considered related to the exogenous inserted gene sequences and its product	- Intrinsic hazardous properties (e.g., toxic, allergenic, oncogenic properties) - Impact on host range, cellular or tissue tropism (biodistribution), shedding, especially when changes on the surface of the virion are expected due to the genetic modification(s) - Impact on replication efficiency - Probability of recombination due to homology with e.g., circulating coronaviruses

## Data Availability

Data sharing not applicable.

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
