# Peer review of "Environmental Risk Assessment of Recombinant Viral Vector Vaccines against SARS-Cov-2"

_vaccines, 2021, doi:10.3390/vaccines9050453_

Round 1
Reviewer 1 Report
The authors propose a review which call attention to the main characteristics of different vaccines against SARS-CoV-2 based on recombinant viral vectors in the pipeline regarding to an Environmental Risk Assessment (ERA) point of view. This is a subject of topical and urgent importance in the present and future context of the COVID-19 pandemic, which is worth to be addressed trying to examine the involved aspects in as much detail as possible.
The proposed review is sufficiently complete and clearly written. However, in the opinion of this reviewer some changes could further ameliorate its completeness and readability.
Here is the list of the suggested changes.
- General. An additional risk on health that could be associated with viral vector based anti-SARS-CoV-2 vaccines is the possibility that virus nucleic acid or protein potentially expressed in specific districts (nasopharynx) of individuals exposed to vaccine materials could interfere with COVID-19 testing. Authors should address this issue somewhere in the review (assessment of the characteristics of the inserted gene sequences??), also distinguishing how this problem is absent or could be overcome in different viral vectors.
- In the “abstract”, line 7, the statement “that are the most advanced” seems not appropriate and should be simply omitted.
- Page 3, at the end of the page, before referring for the first time to table 2, authors should briefly clarify that this table is not directly obtained from official documents of regulatory agencies, such as the EMA guidelines for ERA (ref.21), but is, rather, their subjective concise summary of how ERA could be applied to the specific subject of this review.
- Page 4, line 11, and page 11 line 46, “survive” and “surviving”, better “to retain their infectivity” and “retaining its infectivity”, respectively.
Author Response
We would like to thank the reviewer for critical reading of the manuscript and for his/her positive and constructive comments. In addition to providing an authors’ rebuttal to his/her general suggestion, we further refined the manuscript by taking into account each of the reviewer’s additional suggested changes.
Rebuttal to general suggestion to discuss interference of COVID-19 vaccines with COVID-19 testing
It cannot be excluded that in the particular case of COVID-19 vaccines, which harbour nucleic acid or encode proteins that are also the target of COVID-19 test assays, interference may occur. This might be the case for intranasal administered vaccines or when mucosal membranes of persons (other than vaccinees) may be exposed to COVID-19 vaccines. However, considering that most commonly used RT-PCR assays target multiple SARS-CoV-2 genes that are different from the Spike protein and that most of the recombinant viral vectors discussed in the paper, as well as for example mRNA vaccines, harbour sequences encoding for the protein S, interference with COVID-19 testing may be limited, if not existing at all. Another aspect to consider is that many of the COVID-19 vaccines are non-replicating. This means that even when the nucleic acids, which are targeted by a diagnostic test, are present, these will be in amounts that is several orders of magnitude lower as compared with a SARS-CoV-2 virus infection. Considering all of these elements, vaccination is not likely to interfere with a positive diagnostic test.
The authors are of the opinion that the potential issue on interference with COVID-19 testing does not constitute a risk for the general population as such. Instead, if interference occurs, it merely reveals an issue related to the management of the pandemic at large. Because this issue clearly goes beyond the environmental risk assessment and the scope of the review, we propose not to address this aspect in this review.
Revision in manuscript following the reviewer’s additional suggested changes:
- In the “abstract”, line 16, the statement “that are the most advanced” has been omitted as requested.
- Page 3, with regards clarification on content of Table 2, the following sentence has been added lines 79-81: “For the purpose of this paper, Table 2 gives a concise summary of how ERA could be applied to viral vector based vaccines against SARS-CoV-2” without any reference to official documents of regulatory agencies”.
- Page 4, line 95, and page 11, line 510 : “survive” and “surviving” have been replaced by “to retain their infectivity” and “retaining its infectivity” as requested.
Reviewer 2 Report
Clearly, the work is appreciated and the authors have collected large amounts of (partially) relevant information. Compared to other reports on viral vectors as vaccination platform (e.g., see also influenza, etc.), which are around for decades, I though miss the novelty and relevance concerning the main topic: 'The present article highlights the main characteristics of recombinant viral vector vaccine (candidates) against SARS-CoV-2 in the pipeline and discusses their features from an environmental risk point of view.'
Thus, I feel that the report in its current form does not meet the requirements for publication in Vaccine and suggest that the authors' may consider investing extra work to improve the quality concerning: 'The present article highlights the main characteristics of recombinant viral vector vaccine (candidates) against SARS-CoV-2 in the pipeline and discusses their features from an environmental risk point of view.'
Based on the outcome of these improvements, the authors' may consider resubmission to Vaccines for reevaluation.
Author Response
We would like to thank the reviewer for his/her appreciation of our work.
With regards to the general comment encouraging the authors to improve the quality of the review from an environmental risk point of view, the authors lack further clarification or reference to passages in the manuscript in order to adequately address the reviewers comment.
Nevertheless, the authors take this opportunity to clarify the overall aim and structure of the review.
Several papers reviewed the advent of the use of recombinant viral vectors as vaccines against infectious diseases, including SARS-CoV-2. However, the particularity of this review relies in addressing the environmental risk assessment (ERA) of several recombinant viral vector COVID-19 vaccine candidates at various stages of clinical development or already approved for marketing in the EU. The ERA of GMO vaccines is embedded in a regulatory framework, which is briefly introduced in the introduction, and follows a defined assessment methodology, which is elaborated in Section 2 of the review paper. These aspects are presented in the manuscript to inform and sensitize the reader on the ERA point of view, which is clearly a different objective as compared to the perspective taken in several other review papers on recombinant viral vectors used as COVID-19 vaccine candidate. Indeed, the latter review reports often address the safety for the vaccinee, the quality and/or the efficacy of the vaccine and do not tackle issues that are relevant for the ERA point of view even though some aspects of patient safety assessment and ERA are similar.
With section 3 the authors have chosen to elaborate a non-exhaustive list of recombinant viral vectors which are particularly relevant from an environmental risk point of view. Various aspects related to potential risk for human health and the environment at large are addressed. To increase the readability, the authors created a concise summary of the aspects that have been discussed throughout the afore-mentioned recombinant viral vectors.
In the discussion, the authors share thoughts on the pro and contra’s associated to the temporary derogation of clinical trials using afore-mentioned recombinant viral vector COVID-19 vaccine candidates from the relevant European legislation due to the pandemic.
Considering all of the elements above, the authors think that the review provides relevant tools to conduct an ERA of a wide range of recombinant viral vectors.
Reviewer 3 Report
The manuscript "Environmental Risk Assessment of Recombinant Viral Vector Vaccines against SARS-CoV-2" is a well-organized and well-written review providing wide-ranging literature information on a very interesting topic, i.e., on recombinant viral vectors that are being investigated as candidates or are currently being used for the development of vaccines against SARS-CoV-2. Though there is a temporary derogation from the relevant European legislation due to the pandemic, the above recombinant viral vectors should be considered and handled as genetically modified organisms (GMOs). There is, therefore, an environmental risk assessment point of view concerning use and oversight regulation of these vectors: in other words, in addition to their efficacy and safety for vaccinees (which has been the objective of many research studies and review reports so far), there is also need to assess various aspects related with potential risks for human health in general and the environment at large which the aforementioned vectors might cause. Such aspects are extensively presented in the manuscript under review, providing up-to-date information and sensitizing the reader; in this respect, the manuscript deserves publication in Vaccines.
Minor comments:
p. 1, 2nd paragraph (and p. 8, 4th paragraph): ...in regards to...
Table 1, 4th row, 2nd "column": ...Companies (instead of Compagnies)
p. 6, end of 4th paragraph: "...in particular due to the fact that these proteins are not expressed on the virion surface" -If the authors agree, this might change into: "...in particular due to the fact that these proteins are not expressed on the virion susface, as it happens in other types of COVID-19 vaccine candidates".
p. 6, 5th paragraph: [45], instead of [45 Zahn et al 2012].
p. 10, 2nd paragraph: ...influenza-based vectors are likely to induce an immune response against backbone associated antigens, such as the as hemagglutinin protein
Ref 15: doi: 10.1038/s41467-020-19829 (instead of: doi: 10.1038/s41467-0200-19819)
Ref. 81: ...Immunogenicity, safety, and tolerability of a recombinant measles virus-based chikungunya vaccine: a randomized, double-blind, placebo-controlled, active comparator, first in man trial... (instead of: ...Immunogenicity, safety, and tolerability of a recombinant measles virus-based chikungunya vaccine: an observer and subject blinded, block randomized, active and placebo-controlled first in man trial...)
Ref. 83: Lancet 2019 (instead of Lancet 2018)
Author Response
We wish to thank Reviewer 3 for the positive comments and the in-depth analysis of our paper. With respect to reviewer’s minor comments hereunder, all of them have been implemented in the revised manuscript.
- Page 1, line 40 and page 8, line 320: “with regards to” has been replaced by “in regards to” as requested.
- Table 1, 4th row, 2nd "column": the typo has been corrected as requested: Companies instead of Compag
- Page 6, lines 223-224: the suggested clarification has been added: "...in particular due to the fact that these proteins are not expressed on the virion surface, as it happens in other types of COVID-19 vaccine candidates".
- Page 6, line 229: “Zahn et al 2012" has been deleted.
- Page 10, lines 422-423, the suggested corrections have been made: “...influenza-based vectors are likely to induce an immune response against backbone associated antigens, such as the as hemagglutinin protein”.
- Reference 15, line 759: the typo has been corrected as requested: doi: 10.1038/s41467-020-19829 (instead of: doi: 10.1038/s41467-0200-19819).
- Reference 81, lines 936-938: the corrections has been made as requested: “...Immunogenicity, safety, and tolerability of a recombinant measles virus-based chikungunya vaccine: a randomized, double-blind, placebo-controlled, active comparator, first in man trial...” (instead of: ...Immunogenicity, safety, and tolerability of a recombinant measles virus-based chikungunya vaccine: an observer and subject blinded, block randomized, active and placebo-controlled first in man trial...).
- Reference 83, line 944: the typo has been corrected: Lancet 2019 (instead of Lancet 2018).